# Education and Training of Non-Genetics Providers on the Return of Genome Sequencing Results in a NICU Setting

**DOI:** 10.3390/jpm12030405

**Published:** 2022-03-05

**Authors:** Kelly M. East, Meagan E. Cochran, Whitley V. Kelley, Veronica Greve, Candice R. Finnila, Tanner Coleman, Mikayla Jennings, Latonya Alexander, Elizabeth J. Rahn, Maria I. Danila, Greg Barsh, Bruce Korf, Greg Cooper

**Affiliations:** 1HudsonAlpha Institute for Biotechnology, Huntsville, AL 35806, USA; mcochran@hudsonalpha.org (M.E.C.); wkelley@hudsonalpha.org (W.V.K.); vegreve@med.umich.edu (V.G.); cfinnila@hudsonalpha.org (C.R.F.); tcoleman@hudsonalpha.org (T.C.); mjennings@hudsonalpha.org (M.J.); latonyaa99@gmail.com (L.A.); gbarsh@hudsonalpha.org (G.B.); gcooper@hudsonalpha.org (G.C.); 2Division of Clinical Immunology/Rheumatology, University of Alabama at Birmingham, Birmingham, AL 35294, USA; elizabethrahn@uabmc.edu (E.J.R.); mdanila@uabmc.edu (M.I.D.); 3Department of Genetics, University of Alabama at Birmingham, Birmingham, AL 35294, USA; bkorf@uabmc.edu

**Keywords:** genetics, genome sequencing, provider education, return of results

## Abstract

To meet current and expected future demand for genome sequencing in the neonatal intensive care unit (NICU), adjustments to traditional service delivery models are necessary. Effective programs for the training of non-genetics providers (NGPs) may address the known barriers to providing genetic services including limited genetics knowledge and lack of confidence. The SouthSeq project aims to use genome sequencing to make genomic diagnoses in the neonatal period and evaluate a scalable approach to delivering genome sequencing results to populations with limited access to genetics professionals. Thirty-three SouthSeq NGPs participated in a live, interactive training intervention and completed surveys before and after participation. Here, we describe the protocol for the provider training intervention utilized in the SouthSeq study and the associated impact on NGP knowledge and confidence in reviewing, interpreting, and using genome sequencing results. Participation in the live training intervention led to an increased level of confidence in critical skills needed for real-world implementation of genome sequencing. Providers reported a significant increase in confidence level in their ability to review, understand, and use genome sequencing result reports to guide patient care. Reported barriers to implementation of genome sequencing in a NICU setting included test cost, lack of insurance coverage, and turn around time. As implementation of genome sequencing in this setting progresses, effective education of NGPs is critical to provide access to high-quality and timely genomic medicine care.

## 1. Introduction

Genetic diseases are one of the leading causes of infant morbidity and mortality in the neonatal intensive care unit (NICU) [1,2]. The time it takes for an infant to receive a genetic diagnosis is often far too long to appropriately guide clinical management, highlighting the need for new advances in diagnostic technologies [3]. Multiple recent studies have established the clinical utility of genome sequencing for neonates with suspected genetic disorders, leading to increased diagnostic yield and decreased overall healthcare spending [3,4,5,6,7,8]. Despite proven clinical utility of genome sequencing in the NICU, the genetics workforce, comprised of medical geneticists and genetic counselors, is insufficient to meet current demand for genetic testing in general, especially in the southern United States [9,10,11], and increased usage of genetic testing will exacerbate this problem.

In response to this growing demand, the ordering and interpreting of genetic tests are increasingly provided by non-genetics providers (NGPs), who often have limited genetics knowledge in part due to limited genetics coursework offered during medical training, but also shaped by the rapidly evolving genetics landscape [12,13,14,15]. This gap in knowledge can lead to adverse medical, psychological, and financial events for patients due to inaccurate ordering, misinterpretation of results, or inadequate genetic counseling [16,17,18]. Neonatologists themselves report concerns about genome sequencing regarding the interpretation of results, parental consent, clinical utility of the results, and potential harms of genomic testing [19].

To meet current and expected future demand for genome sequencing in the NICU, adjustments to traditional service delivery models are necessary. These new models may include genetics professionals working in partnership with NGPs, genetics professionals providing consultative services, asynchronous oversight by genetics professionals, or the training of NGPs to effectively provide genetic services [20,21]. Effective programs for the training of NGPs have the opportunity to address known barriers to providing genetic services, including limited genetics knowledge and lack of confidence [22,23,24]. Previous educational interventions have been shown to increase NGP knowledge of genetics and confidence, suggesting that this type of intervention can potentially address demand for genetic services [25,26,27,28].

SouthSeq is a Clinical Sequencing Evidence-Generating Research (CSER) Consortium project exploring the use and impact of genome sequencing in NICU patients in the southeastern US [6]. A diverse population of newborn patients with suspected genetic conditions and their families were recruited from participating clinical sites in Alabama, Mississippi, Louisiana, and Kentucky. SouthSeq aims to use genome sequencing to make genomic diagnoses in the neonatal period and evaluate a scalable approach to delivering genome sequencing results to populations with limited access to genetics professionals.

## 2. Materials and Methods

### 2.1. SouthSeq Study Protocol and Purposes

Recruitment and informed consent of SouthSeq patient participants were facilitated by research nurses at each participating NICU site. Genome sequencing was performed on newborn proband samples, with Sanger confirmation of variants of interest using available parental samples [6]. Primary results (pathogenic, likely pathogenic, and uncertain) related to the newborn’s symptoms were identified and reported. Secondary results (pathogenic and likely pathogenic variants in an actionable gene list) and incidental results were also reported if the participant family consented for their return [29].

When genome sequencing and analysis were complete, results were disclosed to participant families by a study-associated healthcare provider, either in person or via telephone. Genome sequencing results were placed in the newborn’s medical record. If providers were unable to contact a participant family, certified letters were sent to notify families of result availability. A primary aim of SouthSeq is to evaluate whether genome sequencing results can be effectively communicated to patient families in the NICU setting by NGPs. SouthSeq was designed as a non-inferiority trial and participant families were randomized to either receive their genome sequencing results disclosure from a genetic counselor or a trained NICU NGP. An electronic platform, Genome Gateway (HudsonAlpha, Huntsville, AL), was utilized to communicate, provide education, and share documents with providers and participants as well as allow for digital survey completion. SouthSeq trial outcomes include parental empowerment, parental perception of uncertainty, and parental personal utility, as well as monitoring of results disclosure audio recordings for provider errors. This manuscript describes the NGP education protocol and associated outcomes of provider training. Analysis of other clinical trial data is ongoing and will be published elsewhere.

The review board at the University of Alabama at Birmingham (IRB-300000328, date of approval: 29 September 2017) approved and monitored the SouthSeq study, including the provider training intervention. All study participants were required to give written consent to participate in this study.

### 2.2. Study Population

NGPs eligible for SouthSeq participation and the associated training were physicians and mid-level providers (nurse practitioners and physician assistants) working within the NICUs at five participating hospitals across Alabama, Mississippi, Louisiana, and Kentucky (Table 1). Providers either self-selected or were selected by department leadership at their institution to participate in the SouthSeq study.

### 2.3. Training Protocol and Objectives

The NGP training intervention was developed by a team of genetic counselors, utilizing case-based scenarios and focusing on the specific knowledge and skills needed to effectively disclose genome sequencing results. Training content and materials were developed through an iterative process, gathering feedback from members on the study team with expertise in neonatology, genetics, education, and clinical trial design. Participating NGPs were required to attend a live training session lasting approximately four hours. Training events occurred at each clinical site prior to the study launch. The training incorporated a series of brief didactic presentations, hands-on activities, and small group discussions.

Didactic topics included genome sequencing technology, SouthSeq clinical trial logistics, and psychosocial considerations. Hands-on portions of the training used a diverse set of patient and result vignettes to allow trainees to interact with example result reports that represent the variety of result implications possible through genome sequencing. The training intervention culminated with a one-on-one simulation exercise in which NICU providers reviewed and disclosed an example report to a member of the training team. The simulated results disclosure was followed by a debrief discussion between the trainee provider and genetic counselor in which real-time feedback was provided. Comprehensive genetic counseling, interpreting secondary or incidental results, and long-term medical management were considered to be out of the scope of the training intervention and expectations set for participating providers.

Learning objectives included

Explain the benefits and limitations of genome sequencing and how it compares to other types of genetic tests;State the purpose of the SouthSeq study and the hypothesis being tested through result disclosure;Identify the role of the non-genetics NICU provider in the SouthSeq study;Demonstrate familiarity and proficiency completing provider tasks in the online Genome Gateway platform;Interpret a SouthSeq genome sequencing result letter and report;Develop a plan for disclosing various types of genome sequencing results (positive, negative, and uncertain) including key points and next steps;Describe common questions among patients receiving genome sequencing results;Attend to psychosocial needs of families surrounding genome sequencing result disclosure;Identify and critique patient support resources relevant to genome sequencing results.

Training materials and recorded presentations were made available online to participants for asynchronous review throughout the duration of the trial. Remote, virtual training sessions were also conducted as needed to train NGPs unable to attend the in-person training. Providers completing training remotely reviewed the recordings of didactic presentations and participated in a live interaction with a study genetic counselor to complete the discussion and simulation aspects of the training. The training intervention schedule and session descriptions can be found in Appendix A.

In addition to the live training, additional education and resources were provided to NGPs within the actual SouthSeq result reports. Reports were written in language intended to be easily understood by both NGPs and participant families. Report format and verbiage were generated via an iterative process and consultation with experts in health literacy. Reports include a bulleted list of key points about the result including the possible impact of the results on medical care and family members. This “just-in-time” education was specific to a particular patient result and delivered at the time the provider would be using the information to talk with families and guide medical care. An example SouthSeq result report can be found in Appendix A.

### 2.4. Survey Instrumentation

Prior to live training, participating providers completed an online pre-survey. The pre-survey elicited demographic information, current practices regarding genetic and genomic testing, and baseline confidence in understanding genome sequencing results and using genome sequencing results to manage patient care. An online post-survey was completed immediately following the live training intervention. The post-survey included questions regarding the impact of training on increasing relevant knowledge and skills for genome sequencing result disclosure. Response options included a Likert scale of “not at all,” “a little,” “somewhat,” and “very.” The confidence questions from the pre-survey were repeated on the post-survey to measure the change in reported confidence related to participation in the training intervention. The Wilcoxon signed-ranks test was used to analyze pre- and post-confidence levels of matched samples. Finally, the post-survey included a series of open-ended questions for providers to give feedback regarding the most and least useful aspects of training and any additional topics for which they would like to receive education and training. Survey questions were novel and developed by the study team. Survey instruments can be found in Appendix A.

## 3. Results

### 3.1. SouthSeq Non-Genetics Provider Participants

#### 3.1.1. Demographics

A total of 33 neonatology non-genetics providers received training across the 5 clinical sites, including 26 physicians and 7 nurse practitioners. Twenty-seven providers completed the pre-training survey (Table 1). The majority of respondents were white (78%) and early in their career (0–10 years of experience, 52%). More than half (54%) reported no previous formal genetics training. Those who reported previous genetics education described that education as a residency rotation (37%), a genetics course or continuing medical education (CME) (11%), a genetics residency or fellowship (4%), or other experiences (4%). Selection of participating providers varied among clinical sites based on a variety of factors, including provider interest and clinical capacity. Therefore, the demographics of participating providers are not necessarily representative of the larger NICU clinical teams at each clinical site.

#### 3.1.2. Prior Experience with Genetic and Genomic Testing

Prior to SouthSeq training, providers reported that they order genetic tests approximately once per week (33%), once per month (48%), or once per year (19%). The pre-survey elicited self-reported confidence in the abilities to read and interpret genetic test results (i.e., single gene tests and gene panel tests), read and interpret genome sequencing results, and manage a patient’s care based on genome sequencing results. Compared to confidence reading and interpreting genetic test results and managing patients based on genome sequencing results, NGPs expressed the lowest confidence in their ability to read and interpret genome sequencing results; 78% of NGPs reported that they were a little confident or not at all confident in this domain, with no NGPs stating that they were very confident in this area (Figure 1). Further, 66% of respondents reported they had never seen a genome sequencing result in their clinical practice.

The pre-survey also elicited perceptions of barriers to implementation of genome sequencing in the NICU setting. Respondents selected from a pre-defined list of barriers, with the ability to select multiple responses. The pre-defined list was generated by the study team, with an option for respondents to add other barriers not included on the list. The most frequent barriers selected being test cost (81%), lack of insurance coverage (81%), turnaround time (81%), and lack of healthcare provider knowledge/training (56%) (Table 2). In contrast, no respondents indicated that no barriers existed and only 7% cited a lack of diagnostic value as a barrier for implementation of genome sequencing.

### 3.2. Impact of the SouthSeq Training Intervention

#### 3.2.1. Provider Understanding and Skills

Twenty-three providers completed the post-training survey. The post-training survey elicited feedback from participants about the perceived impact of the training intervention. Respondents were asked to select to what extent training increased their understanding of genomics and the role it can play in making a diagnosis. Respondents mostly indicated that the training intervention had a positive impact on this understanding, answering “very” (*n* = 11.48%), “somewhat” (11.48%), and “a little” (1.4%). The survey also asked respondents to select to what extent training equipped them with the knowledge and skills needed to implement the provider role in SouthSeq. Respondents indicated a positive impact of training, with the majority of respondents (18.78%) selecting the “very” option, with the remaining respondents selecting “somewhat.”

#### 3.2.2. Provider Confidence

Following the training intervention, providers were asked to re-evaluate their perceived confidence in their ability to read and interpret genome sequencing results (Figure 2), as well as confidence in managing a patient’s care based on genome sequencing results (Figure 3). Confidence ratings increased in both domains after participating in SouthSeq provider training. The post-test median response for each of these questions was “somewhat confident,” a full response category higher than the pre-test median response. A Wilcoxon signed-ranks test indicated that the median post-test scores were statistically significantly higher than the median pre-test scores Z = 4.035, *p* < 0.001.

A Wilcoxon signed rank test revealed that perceived confidence scores were significantly lower before the intervention (Median = 2, *n* = 23) compared to after (Median = 3, *n* = 23), *z* = −4.035, *p* ≤ 0.001, with a strong effect size, *r* = 0.59 [30].

### 3.3. Qualitative Feedback

The final section of the post-training survey elicited free-text responses regarding the most and least valuable aspects of the training intervention. Eleven respondents (48%) reported that the simulated results disclosure with genetic counselors was the most valuable aspect of training. Other aspects of training mentioned in free-text responses as being most valuable included reviewing example result reports, understanding what genome sequencing does, and being able to ask questions about the study protocol. Most respondents (18/23) did not mention any specific training aspects that were least valuable. However, a minority of respondents mentioned they found the didactic portions of training to be least valuable. Four respondents provided feedback on additional topics they wish had been covered, or covered in more depth, as a part of training. These topics included more information about incidental findings, different types of genetic testing, and genome sequencing.

## 4. Discussion

The SouthSeq study aimed to explore the diagnostic utility of genome sequencing in a diverse population of patients in a NICU setting and test a scalable approach to the implementation of genome sequencing. SouthSeq and other related studies have established the clinical utility of genome sequencing for neonates with suspected genetic disorders, leading to increased diagnostic yield and decreasing overall healthcare spending [3,4,5,6]. However, current workforce shortages of medical geneticists and genetic counselors, the skewed geographic distribution of genetics providers, and limited genetics knowledge among NGPs hinder the widespread implementation of genome sequencing in the NICU setting [9,10,11,23]. To aid in addressing these barriers, the SouthSeq study tested a scalable model of genome sequencing results disclosure by trained non-genetics providers. As a part of this model, neonatology NGPs received live, interactive training by a team of genetic counselors and were provided with enduring educational resources.

Participation in the live training intervention led to an increased level of confidence in critical skills for the clinical implementation of genome sequencing. Providers reported a significant increase in confidence level in their ability to review, understand, and use genome sequencing result reports to guide patient care. Providers also reported gains in knowledge regarding the use of genomics in making diagnoses and the role of the NGP within the SouthSeq project specifically. These outcomes, while unique in technology used and clinical context, are in line with the outcomes observed in other successful NGP genetics training interventions [25,26].

When asked to evaluate the most useful components of the training protocol, nearly half of participants identified the value of the interactive and practical components, including simulated results disclosures and reviewing example reports, while a small minority found didactic components to be least valuable. Prior studies have indicated that problem-based learning and interactive learning interventions such as the one described in this manuscript are effective in improving the knowledge of genetics and clinical skills of NGPs and are training modalities that are favored by NGPs [26,27,31]. Although studies suggest that these forms of continuing medical education (CME) are effective, substantial barriers remain to widespread adoption including provider time, financial considerations, preference for other methods or content of CME, lack of awareness of genetics CME, and the geographical distance of CME offerings from provider practice location [32,33]. Additionally, genetics education interventions such as the one explored in this study require a substantial investment of time and resources by genetics experts to develop content, create resources, and facilitate training sessions.

Despite a growing body of evidence demonstrating clinical utility, genome sequencing remains an infrequently used diagnostic test in NICUs [34]. Data presented here provide insight into perceived barriers among NICU NGPs regarding the implementation of genome sequencing. Few surveyed providers reported lack of diagnostic value as a barrier. A larger minority of respondents cited unexpected and uncertain results as an implementation barrier. These findings are similar to the NICU provider perspectives identified in other studies [19]. Logistical issues including cost, lack of insurance coverage, and turnaround time were reported to be barriers by most respondents. These logistical barriers may be overcome in the future as genome sequencing technology improves and the evidence for clinical utility continues to grow. More than half of respondents selected a lack of provider knowledge and training as a barrier to genome sequencing implementation, a finding in line with other publications [35].

The training intervention described in this study and its preliminary outcomes demonstrate the value of genetic counselors and other genetics experts as educators of NGPs, a role echoed in genetic counselor practice-based competencies and often reported as a component of job duties [36,37]. Due to limited access to genetics specialists, adjustments to traditional service delivery models are critical, which may include educating NGPs to provide genetic services as in the SouthSeq study or other models such as genetics professionals working in tandem with NGPs or genetics professionals providing consultative services [20]. In the future, additional data surrounding the outcomes of disclosure of genome sequencing results by NGPs in the SouthSeq study will be made available, including data on the frequency and nature of errors in results disclosure and comparisons to disclosure of results by genetic counselors. Further research is needed to evaluate the outcomes of the implementation of this novel service delivery model and its ability to provide access to accurate, effective, and timely genetic testing and counseling services.

The interpretation of our study findings are limited by a relatively small sample of NGPs. Due to the limited sample size, we were unable to assess the effect of provider demographic variables on outcome data. The training intervention described and tested would benefit from additional study in a larger population of NGPs and in additional clinical contexts beyond the NICU. Opportunities for future research include assessing whether the increased levels of provider confidence observed post-training are sustained over time and whether participation in SouthSeq had an impact on the use of genome sequencing by participating providers and institutions, s well as other measures of objective impact. Measuring outcomes based on perceived confidence is limited by its subjective and potentially transient nature, particularly in the setting of acutely ill infants.

Herein, we describe the protocol for a live, interactive educational intervention utilized in the SouthSeq study and the associated impact on NGP knowledge and confidence in reading, interpreting, and using genome sequencing results. As the body of evidence for the diagnostic utility of genome sequencing in critically ill newborns grows, utilization of genome sequencing is expected to increase, placing a higher demand on NGPs to facilitate this testing. As implementation of genome sequencing in this setting progresses, effective education of NGPs is critical to provide access to high-quality and timely genomic medicine care.

## Figures and Tables

**Figure 1 jpm-12-00405-f001:**
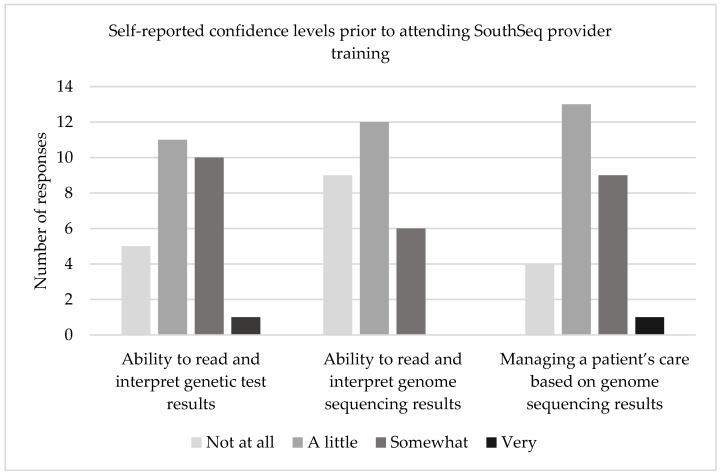
Baseline, self-reported confidence levels prior to attending SouthSeq provider training intervention related to ability to read and interpret genetic test results, ability to read and interpret genome sequencing results, and managing a patient’s care based on genome sequencing results.

**Figure 2 jpm-12-00405-f002:**
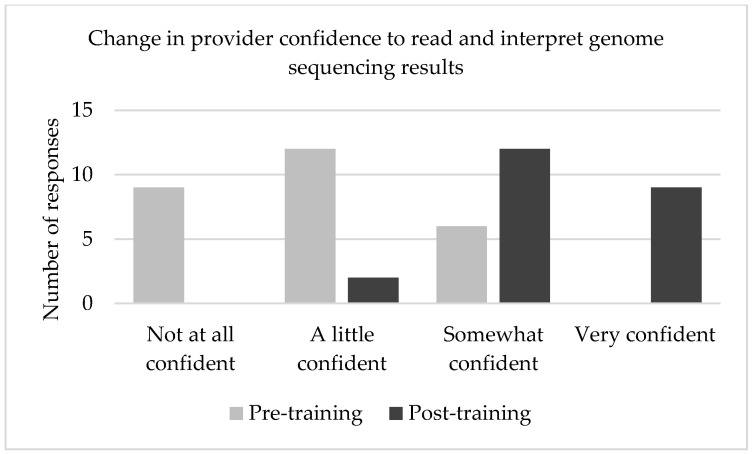
Comparison of self-reported provider confidence before and after participating in SouthSeq training about ability to read and interpret genome sequencing results.

**Figure 3 jpm-12-00405-f003:**
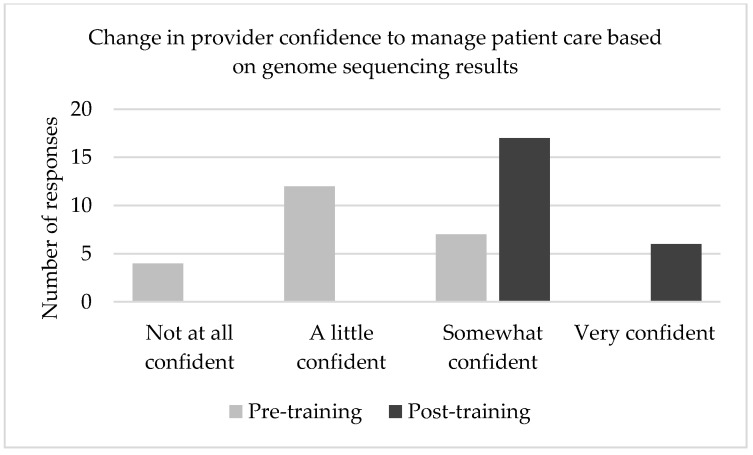
Comparison of self-reported provider confidence before and after participating in SouthSeq training about ability to manage a patient’s care based on genome sequencing results.

**Table 1 jpm-12-00405-t001:** Demographic information for non-genetics neonatology providers participating in SouthSeq.

Clinical Site	Frequency (%)
University of Louisville, Louisville, KY, USA	10 (31%)
Woman’s Hospital, Baton Rouge, LA, USA	8 (24%)
University of Alabama at Birmingham, Birmingham, AL, USA	7 (21%)
University of Mississippi Medical Center, Jackson, MS, USA	5 (15%)
Children’s Hospital, New Orleans, LA, USA	3 (9%)
**Race**	**Frequency (%)**
White	21 (78%)
Black	4 (15%)
Asian	2 (7%)
**Years of Experience**	**Frequency (%)**
0–5 years	10 (37%)
6–10 years	4 (15%)
11–15 years	3 (11%)
16–20 years	4 (15%)
21–25 years	2 (7%)
25+ years	4 (15%)

**Table 2 jpm-12-00405-t002:** Perceived barriers to implementation of genome sequencing in the NICU setting.

Barrier	*n* (%)
Test cost	22 (81%)
Lack of insurance coverage	22 (81%)
Turnaround time	22 (81%)
Lack of healthcare provider knowledge/training	15 (56%)
Possibility of uncertain results	13 (48%)
Possibility of unexpected results	9 (33%)
Limited healthcare provider time	3 (11%)
Limited diagnostic value	2 (7%)
Other	0 (0%)
There are no barriers	0 (0%)

## Data Availability

The data presented in this study are available in Appendix A.

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
