# Peer review of "Education and Training of Non-Genetics Providers on the Return of Genome Sequencing Results in a NICU Setting"

_jpm, 2022, doi:10.3390/jpm12030405_

Round 1

Reviewer 1 Report

  1. Please edit the abstract as it should reflect the whole manuscript
  2. Please describe the genetic diseases.
  3. Please describe the technique in detail
  4. Compare the study results with other diseases in the global studies.
  5. Add the meta-analysis studies if applicable.

Author Response

Point 1: Please edit the abstract as it should reflect the whole manuscript

Response 1: Additional details were added to the abstract to specify the sample size and to include findings regarding percieved barriers.

Point 2: Please describe the genetic diseases.

Response 2: The specific genetic diseases diagnosed through the SouthSeq study are not in scope of the current paper. The genomic findings of SouthSeq have been previously published in reference number 6. We have added an additional reference to this paper in text line 72 to point readers to this manuscript if interested in learnng more aboout SouthSeq genetic disease data.

Point 3: Please describe the technique in detail

Response 3: Technique regarding genome sequencing and genomic results are detailed in a separate publication (reference 6). We have added additional information in text regarding the technique used to develoop and implement the SouthSeq provider education intervention, and reorganized that section of the manuscript for clarity. Specifically, we have added a supplementary material file to the re-submission that includes a draft SouthSeq report. This draft report includes an example of the result-specific “just-in-time” education provided and technical inforamtion about the testing assay.

Point 4: Compare the study results with other diseases in the global studies.

Response 4: Outcomes of our study are compared to other studies within and outside the US at several points throughout the discussion (lines 321-322, 326-328, 347-348; references 25-28, 31). Specifically, perceived knowledge gains and positive feedback received regarding the interactive components of training are in line with findings regarding the effectiveness of problem-based and interactive learning in other studies. Further, findings regarding barriers to genome sequencing implementation are also compared to other studies. Additionoal context has been added in the discussion in line 325-326 to highlight the uniqueness of our study in comparison to other studies that have seen similar outcomes.

Point 5: Add the meta-analysis studies if applicable.

Response 5: There is a notable paucity of meta-analysis in the field of non-genetics provider education in genetics and genomics, in part due to a lack of standards for reporting on genomics education and evaluation (Niselle, A, et al, 2021, PMID 33824503).

Reviewer 2 Report

Dear authors,

thank you for your manuscript. While this manuscript is well written, it would benefit from more statistical analyses of the data at hand:

  • Is there a group difference of change in provider confidence on the basis of clinical site, race, role (physician vs nurse practitioner) or years of experience?
  • Is there effect of years of experience or role on perceived barriers?
  • How does the characterization of this cohort change daily robustness of provider confidence?

Please give full results of statistical analyses according to APA format wherever needed.

Author Response

Point 1: Is there a group difference of change in provider confidence on the basis of clinical site, race, role (physician vs nurse practitioner) or years of experience?

Response 1: We agree that a sub-analysis on provider confidence change based on provider demographic variables would be hugely beneficial. However our relatively limited sample size did not allow for these sub-analyses to be done in a statistically meaningful way. We have expanded our discussion of study limitations to include this in lines 369-377. This type of training intervention would benefit from further study in a larger populatioon of non-genetics providers.

Point 2: Is there effect of years of experience or role on perceived barriers?

Response 2: We were unable to analyze the effect of provider years of experience on study outcomes due to the limited sample size. This is an important question that would benefit from additional study in the future.

Point 3: How does the characterization of this cohort change daily robustness of provider confidence?

Response 3: We agree that measuring provider confidence is limited in robustness given its subjective and possibly transient nature. The length of the training intervention and the timing of survey assessments allowed limited opportunity for other variables to interact with a provider’s perceived conficence level beyond the training inervention. However, we acknowledge the limitations present using provider confidence as an outcome measure, particularly in the setting of acutely ill infants. We have added additional discussion around this limiation in the discussion section of the manuscript in lines 375-377.

Point 4: Please give full results of statistical analyses according to APA format wherever needed.

Response 4: We have added additional statistical detail regarding the Wilcoxon signed rank test results, including the effect size, in line with APA formatting in lines 278-280. 

Reviewer 3 Report

This is a well and clearly structured article describing a worth-while training programme. However, the authors cited several previous NGP training interventions (references 25,26) which have reported similar outcomes. Therefore one could ask what is value in reporting the outcome of this study. The reviewer would encourage the authors to add more details on how this study sets itself apart from other similar studies reported in the past.

There are a few other minor suggestions:

  1. (Line 75) ‘SouthSeq results were placed in the newborn’s medical record.’ It is unclear what is meant by SouthSeq results.
  2. Table 1 states the demographic information for participants including their race. Here it might also be beneficial to state the male/female frequency. Also, Black and Asian minorities seem to be underrepresented.
  3. 2.3 Training Protocol and Objectives: Was the training specific to different types of genetic diseases? The implications and significance of genomic sequencing results can differ immensely depending on the affected gene.
  4. (Line 139). Please define ‘just-in-time resources’.
  5. Figure 1: ‘Ability to read and interpret genetic test results’ versus ‘Ability to read and interpret genome sequencing results’. Here it is not clear what the difference is as the article only refers to genome sequencing results. Please define the difference between a genetic test result and a genome sequencing result.
  6. (Line 188): Who created the ‘Pre-defined list of barriers’?
  7. 3.3.2. Provider Confidence. How were confidence levels measured? Were there pre-determined markers of confidence or was it based on self-perception?

The authors have done well in acknowledging the barriers which are currently still encountered with regards to advancing NGP training programmes.

Author Response

Point 1: This is a well and clearly structured article describing a worth-while training programme. However, the authors cited several previous NGP training interventions (references 25,26) which have reported similar outcomes. Therefore one could ask what is value in reporting the outcome of this study. The reviewer would encourage the authors to add more details on how this study sets itself apart from other similar studies reported in the past.

Response 1: We really appreciate this feedback. We have added additional conext to the discussion highlighting that while the outcomes are similar to other studies, that our study is unique in both technology and clinical context.

Point 2: (Line 75) ‘SouthSeq results were placed in the newborn’s medical record.’ It is unclear what is meant by SouthSeq results.

Response 2: We have changed this terminology to be “genome sequencing” results for clarity.

Point 3: Table 1 states the demographic information for participants including their race. Here it might also be beneficial to state the male/female frequency. Also, Black and Asian minorities seem to be underrepresented.

Response 3: This is a great suggestion. Unfortunately, we did not systematically capture reported male/female frequency in our survey like we did other demographic variables included in Table 1 and therefore cannot report that data. Black and Asian minorities are underrepresented in the provider population. We have added additional inforamtion in-text within the result section about demographics about how participating providers were selected and an acknowledgement that the demographics of providers may not be representative of the larger clinical team at each site in lines 227-230.

Point 4: 2.3 Training Protocol and Objectives: Was the training specific to different types of genetic diseases? The implications and significance of genomic sequencing results can differ immensely depending on the affected gene.

Response 4: The implications and significance of genome sequencing results can differ widely depending on the gene and disease implicated. The training intervention was not specific to a set of diagnoses but rather covered the spectrum of possible results using case examples (ex. positive dominant result, positive recessive result, uncertain result, seconodary finding, etc). We have added additional clarification about this in-text in lines 115-116.

Point 5: (Line 139). Please define ‘just-in-time resources’.

Response 5: We have added more information in-text about what we are referring to as “just-in”time resources on the result reports in lines 159-167. We have also added an additional supplementary file that includes an example result report for readers to reference.

Point 6: Figure 1: ‘Ability to read and interpret genetic test results’ versus ‘Ability to read and interpret genome sequencing results’. Here it is not clear what the difference is as the article only refers to genome sequencing results. Please define the difference between a genetic test result and a genome sequencing result.

Response 6: The phrase genetic test results is meant to include genetic tests that analyze one or more genes but is not comprehensive to looking across the entire genome, such as single gene tests or gene panel tests. We have added a definition alongside the phrase genetic test result in line 235.

Point 7: (Line 188): Who created the ‘Pre-defined list of barriers’?

Response 7: The study team created the pre-defined list of barriers included on the survey instrument, and included a free-text other answer to allow respondents to add barriers that were not included ono the pre-defined list. We have added additional inforamtion about this in-text in lines 249-250.

Point 7: (Line 188): 3.3.2. Provider Confidence. How were confidence levels measured? Were there pre-determined markers of confidence or was it based on self-perception?

Response 7: Confidence levels were measured based on provider self-perception before and after participating in the training intervention. We acknowledge the limitiations that exist when using self-perceived measures of impact and have added additional discussion of this limitation in text in lines 375-377.

Round 2

Reviewer 2 Report

Dear authors,

thank you for answering the questions. All in all, it is an encourageable project with a novel question that will be of interest to the readers. Since the design and/or sample size does not give a handle on a few questions, it needed to be discussed and you changed the manuscript accordingly. 
I recommend to accept the manuscript. 
Best of luck!